# Gene–gene interaction detection with deep learning

Tianyu Cui [1✉], Khaoula El Mekkaoui[1], Jaakko Reinvall[1], Aki S. Havulinna [2,3], Pekka Marttinen [1,2,5] & Samuel Kaski[1,4,5]

The extent to which genetic interactions affect observed phenotypes is generally unknown because current interaction detection approaches only consider simple interactions between top SNPs of genes. We introduce an open-source framework for increasing the power of interaction detection by considering all SNPs within a selected set of genes and complex interactions between them, beyond only the currently considered multiplicative relationships. In brief, the relation between SNPs and a phenotype is captured by a neural network, and the interactions are quantified by Shapley scores between hidden nodes, which are gene representations that optimally combine information from the corresponding SNPs. Additionally, we design a permutation procedure tailored for neural networks to assess the significance of interactions, which outperformed existing alternatives on simulated datasets with complex interactions, and in a cholesterol study on the UK Biobank it detected nine interactions which replicated on an independent FINRISK dataset.

[1] Department of Computer Science, Aalto University, Espoo, Finland. [2] Finnish Institute for Health and Welfare (THL), Helsinki, Finland. [3] Institute for Molecular Medicine Finland, FIMM-HiLIFE, Helsinki, Finland. [4] Department of Computer Science, University of Manchester, Manchester, UK. [5] These authors contributed equally: Pekka Marttinen, Samuel Kaski. ✉email: tianyu.cui@aalto.fi

Genome-wide Association Studies (GWAS) have successfully identified risk alleles for complex diseases by associating single nucleotide polymorphisms (SNP) with disease-related phenotypes. Despite this, GWASs usually fail to capture statistical epistasis, i.e., interactions between genes, which has been recognized as fundamentally important to understanding the structure and function of genetic pathways. Mapping interactions between genes holds the promise of a breakthrough in revealing biological mechanisms of diseases, assessing individuals' disease risk factors, and developing treatment strategies for precision medicine[1].

Current scalable statistical approaches lack power to reveal pairwise interactions among a set of candidate genes, because they make restrictive assumptions about how genes are represented, and about the form of the interactions. Popular methods represent interactions between genes by interactions between the top SNPs (the SNP most correlated with the phenotype) of the corresponding genes[2]; this ignores the effects of other SNPs within the same gene. Some scalable approaches have been proposed to summarize the multi-dimensional SNPs into a one-dimensional representation by unsupervised dimensionality reduction methods, e.g., PCA[3], but such learned representations neglect information about the phenotype. Moreover, the interactions are modelled with parametric models, and simplifying assumptions are made when choosing the forms of the models. Linear regression with multiplicative interactions[2,4] is a common choice. Methods based on such simplifications will fail when the actual interactions are more complex. Although more flexible machine learning algorithms, such as boosting trees[5], have been proposed to model complex interactions between SNPs, it is still an open question how can we best combine gene representation learning and modelling of the interactions into an end-to-end model to increase the statistical power.

Deep neural networks (NNs), have been successfully applied to numerous tasks in the biomedical domain, such as predicting protein structures[6] and promoters[7], with large amounts of data. With a properly designed architecture, a NN learns multi-level representations by composing simple but non-linear functions that transform low-level features into abstract high-level representations, allowing the model to automatically discover optimal feature representations and approximate non-linear relations between independent and dependent variables[8]. Although NNs are effective in various prediction tasks, their black-box nature limits their usage in applications requiring model interpretability, such as GWAS. With recent progress in interpretable neural networks, interaction effect between features can be obtained by calculating well-principled interaction scores, such as the input Hessians[9] and Shapley interaction scores[10], which provide a way to estimate gene–gene interactions from NNs trained on GWAS datasets.

Assessing the significance of an interaction is equally important to estimating the interaction itself. As the null distribution of the interaction score is often analytically intractable for neural networks, permutation tests[11] are used as a default solution. Such tests have been widely applied when detecting interactions with linear regression[12]. Previous approaches either 1. permute the dependent variable directly[13] or 2. permute residuals after subtracting the null hypothesis (i.e., a linear regression without interactions) from the dependent variable[14], and use this as the regression target when constructing the null distribution for an interaction. Notably, both of these approaches remove the main effect from the regression target. This is appropriate in linear regression where including the main effects will not change the test statistics of interactions[14]. However, unlike in linear regression, in neural networks the main and interaction effect are entangled in the high-dimensional nonlinear representations. If the main effects were excluded during permutation, NNs would learn representations that are very different from those learned from the original dataset, where main effects are included. Alternatively, they might even miss the signal altogether, which would lead to highly biased null distributions for the interaction effect.

In this paper, we present a deep learning framework to detect gene–gene interactions for a given phenotype, as well as a permutation procedure to accurately access the significance of the interactions. Specifically, we design a neural network architecture with structured sparsity that learns gene representations from all SNPs of a gene as hidden nodes in a shallow layer, and then learns complex relations between the genes and phenotypes in deeper layers. Interactions between genes are then estimated by calculating Shapley interaction scores between the hidden nodes that represent the genes. Moreover, we present a permutation test for interactions in NNs, in which the target is the sum of the permuted interaction effect (residual) and the main effect estimated by a main effect NN. This helps NNs to learn gene representations in the permuted datasets that are similar to those in the original data. We demonstrate with simulations that the proposed permutation test outperforms the existing alternatives for NNs, and that the proposed deep learning framework has increased power to detect complex interactions compared to existing methods. We then show that NNs can find from UK Biobank datasets[15] significant interactions that current approaches may ignore, and we solidify our findings by replicating them in an independent FINRISK dataset[16].

## Results

### Proposed methods

*Gene interaction neural network.* A NN is a function mapping inputs to outputs, by composing simple but non-linear functions. Each hidden layer of the NN represents one composition. A $D$-layer NN $f(\mathbb{X})$ with input $\mathbb{X}$ is defined as $f(\mathbb{X}) = f^{(D)}(\ldots f^{(2)}(f^{(1)}(\mathbb{X})))$, where $f^{(i)}(\cdot)$ represents the $i$th layer. In a GWAS, the input $\mathbb{X}$ comprises SNPs and the output $f(\mathbb{X})$ consists of the predicted phenotype for an individual. The architecture of a NN reflects the inductive biases of the model, i.e., preferences to certain kinds of functions[17], and thus affects the learned representations. For example, the convolutional and pooling layers in convolutional NNs learn translation invariant functions[18] that are especially suitable for computer vision tasks. To learn representations of genes from SNPs, we propose a structured sparse NN architecture, in which SNPs from the same gene are connected already in lower layers with fully connected multilayer perceptrons (MLPs), but SNPs from different genes are not connected until after a special hidden layer, here referred to as the 'gene layer', where each hidden node represents a single gene. After the gene layer all nodes become fully connected (using another MLP) and finally predict the phenotype $\mathbf{y}$, as shown in Fig. 1a. MLPs are universal function approximators[19], which are capable of approximating any feature interactions in the data without making any explicit assumptions about their forms. Therefore, the interactions between genes will be implicitly encoded into the NN after training, and we use gene–gene interaction score (introduced below) to reveal interactions from the trained NN. The architecture reflects the domain knowledge[20] about how SNPs affect the phenotype, according to which the effect of SNPs on a phenotype is assumed to follow a two-stage procedure: 1. SNPs within a gene affect how the gene behaves (e.g., through gene expression); 2. The combined behavior of multiple genes affects the phenotype, and gene–gene interactions are estimated from the second stage.

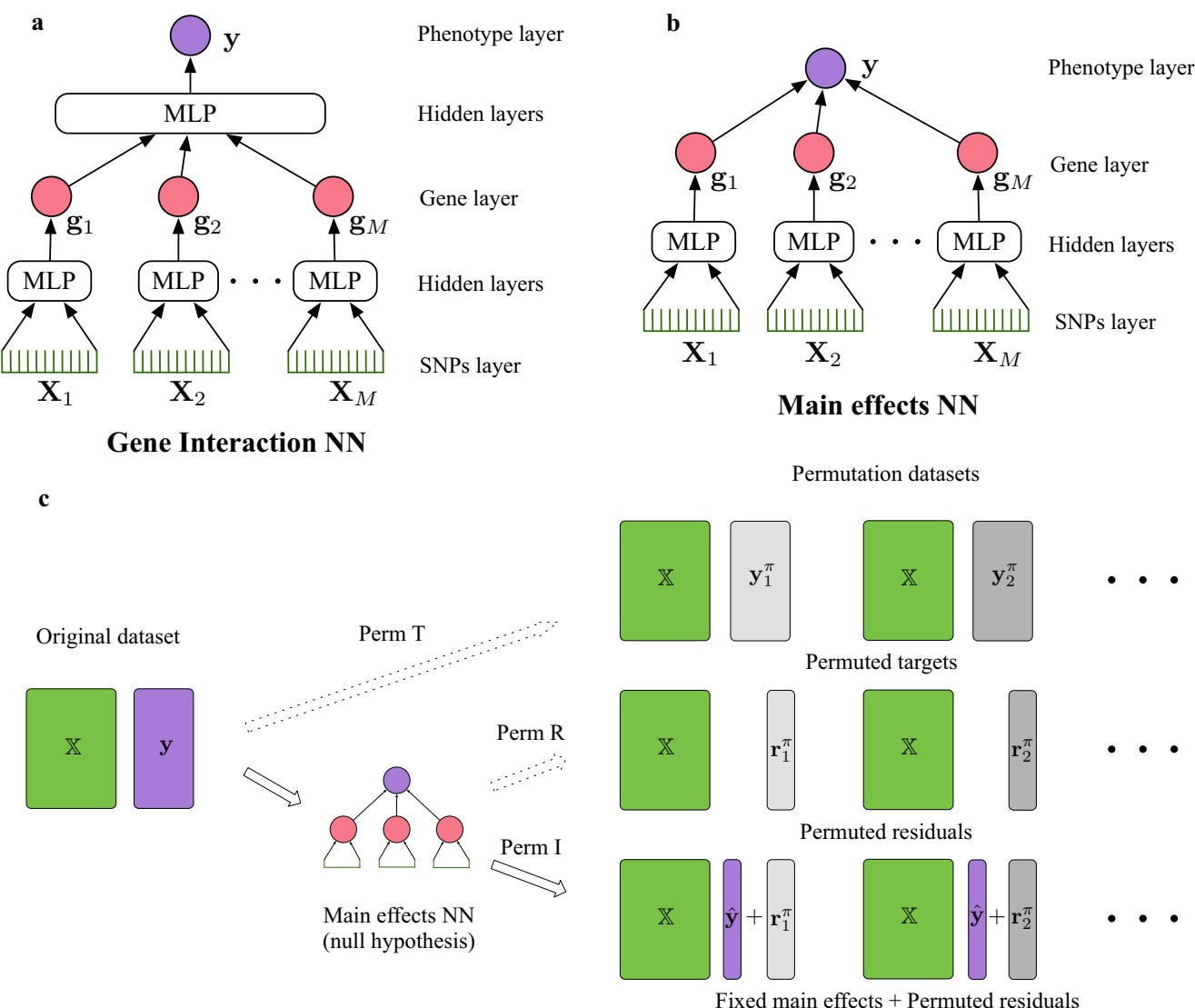

**Gene Interaction NN**

**Main effects NN**

**Permutation Test on Interactions**

**Fig. 1 A framework for detecting gene–gene interaction with deep learning. a** Gene interaction neural network to learn interactions between genes, which are estimated by the Shapley interaction scores between nodes (red) in the gene layer. **b** Main effect neural network (NN) to learn the main effects between genes and used to generate permuted datasets. **c** An overview of the proposed permutation test on interactions. Compared with existing permutation tests (Perm T and Perm R), our proposed permutation procedure (Perm I) includes the estimated main effects of genes from the main effect NN. Different grayscales represent different permutations.

*Gene–gene interaction score.* Shapley interaction score[10] is a well-axiomatized measure of interaction between features that can be applied to any black-box model. To apply it to NNs, we denote the set of all input features by $F$, one feature $i \in F$, and a feature set $S \subseteq F$. The interaction effect between features $i$ and $j$, given a set of features $S$ are presented, of a NN $f$ at data point $X_k$ is given by $\delta_{ij}^f(X_k; S) = f(X_k; S \cup \{i, j\}) - f(X_k; S \cup \{i\}) - f(X_k; S \cup \{j\}) + f(X_k; S)$, where $f(X_k; S)$ is the prediction at $X_k$ when only features in $S$ are used and other features are replaced by their corresponding baseline values (e.g., mean)[21]. The Shapley interaction score is the expectation of $\delta_{ij}^f(X_k; S)$ over $X_k$ and feature sets $S$ sampled uniformly from the power set of $F$. We define the gene–gene interaction score as the Shapley interaction score between hidden nodes in the gene layer, i.e., we consider the learned gene representations as the input features of the Shapley

score. Shapley values are criticized for performing poorly on correlated features[22]. Fortunately, unlike SNPs that are highly correlated, gene representations learned from different SNPs sets are usually relatively uncorrelated (see details in Supplementary Note 1).

*Permutation test for interactions.* The null hypothesis here is that genes only have main effects on the phenotype and no interactions. This we formalize as the 'main effects NN' (Fig. 1b) where only a linear layer is applied after the gene layer. To construct a permuted dataset, we first train the main effects NN on the original dataset and then permute the residual. The dependent variable in the permuted dataset is defined as the sum of the predicted main effect and permuted residual, while the independent variable is the same as in the original dataset (Fig. 1c). To study the calibration of the proposed permutation procedure, we

calculate the false positive rate (FPR), and also consider the maxT permutation[23] for multiple hypothesis correction (i.e., estimate the null distribution for the maximum test scores in permutations) with simulated data. In real-world data experiments, we use the permutation method to calculate the false discovery rate (FDR)[24] to address the multiple testing problem. We refer the reader to the Methods section for more implementation details.

In the rest of this section, we carry out three simulation studies and a cholesterol study with three phenotypes on real-world datasets. In simulation studies, we first simulate complex interactions between genes to validate that 1. the proposed permutation method outperforms existing approaches in estimating the significance of complex interactions, and 2. the proposed neural network method has increased power to detect complex interactions. We then simulate simple interactions between top SNPs, and we notice that the classical top-SNP approaches have a similar or higher power than neural networks to detect simple interactions. Finally, we simulate phenotypes without any interactions to confirm that the proposed permutation procedure yields well-calibrated significance estimates for interactions. In the real-world cholesterol study, we detect gene–gene interactions with the proposed framework in the UK Biobank[15] dataset and replicate the findings in an independent FINRISK[16] dataset.

## Simulation with complex interactions
*Setting.* As the genotypes, we select genes associated with the cholesterol phenotype from the UK Biobank, from which we select all SNPs from 10 genes after basic quality control (see Methods). We simulate the phenotype **y** by first simulating the expression $\mathbf{g}_i$ of each gene $i$ according to a linear combination of all SNPs in the gene, and then generate phenotypes with main and interaction effect (Eq. (1)), as follows:

$$\mathbf{g}_i = \sum_{j=1}^{d_i} \alpha_{ij}\mathbf{x}_{ij}, \quad \forall i \in \{1, \dots, 10\},$$

$$\mathbf{y} = w_{12}\max\{\mathbf{g}_1, \mathbf{g}_2\} + w_{79}\max\{\mathbf{g}_7, \mathbf{g}_9\} + w_{35}(\mathbf{g}_3 - \mathbf{g}_5)^2$$
$$+ w_{48}(\mathbf{g}_4 - \mathbf{g}_8)^2 + w_{67}\mathbf{g}_6\mathbf{g}_7 + w_{810}\mathbf{g}_8\mathbf{g}_{10} + \sum_{k}^{10} w_k\mathbf{g}_k + \epsilon,$$

$$(1)$$

where $\mathbf{x}_{ij}$ is the $j$th SNP of gene $i$. This includes six interactions of three types: multiplicative, $\mathbf{g}_i \times \mathbf{g}_j$, maximum, $\max\{\mathbf{g}_i, \mathbf{g}_j\}$, and (squared) difference, $(\mathbf{g}_i - \mathbf{g}_j)^2$, and we refer to the six interactions in the simulator as 'true' interactions while the other pairwise interactions are 'false'. We adjust the sparsity level of $\alpha_{ij}$ to generate datasets with different proportions of causal SNPs for the simulated phenotypes. Moreover, we set the weights of interactions $w_{ij}$ and main effects $w_k$ according to the variance ratio between the main effect and interaction effect, i.e., main-to-interaction ratio (M/I), and we tune the variance of noise $\epsilon$ to adjust the signal-to-noise ratio (S/N) of the data. All parameters in the simulator, i.e., $\alpha_{ij}, w_{ij}, w_i$, are Gaussian distributed. We consider three different S/N: {0.1, 0.5, 1.0}, which cover realistic GWAS cases, e.g., the S/N of BMI[25] is larger than 1 and HDL[26] has a S/N $\approx$ 0.1, three different data sizes: {40k, 80k, 120k}, three different proportions of causal SNPs: {10%, 50%, 100%}, and three different M/I: {0.25, 1.0, 4.0}. We simulate data using S/N = 0.1, datasize = 80k, prop causal SNPs = 50%, and M/I = 1.0, and we vary one hyper-parameter at a time while fixing others.

We compare the NN framework with six baselines, which are composed by two ways of representing genes (top-SNP and PCA) and three ways of modeling interactions between gene representations i.e., linear regression (LR), Lasso with multiplicative interaction terms, and boosting tree (XGB). We first compare

different permutation methods for each baseline (Fig. 2). For NNs, we compare the proposed permutation method, i.e., using permuted interactions (Perm I), with 2 existing approaches: using permuted target (Perm T) and permuted residual (Perm R). For Lasso and XGB, we compare two existing approaches: Perm T and Perm R, as Perm I is designed for NNs specifically. We do not use any permutation for LR whose null distribution is analytically tractable. We then compare the accuracy of interaction detection, measured by the average precision (AP) from the precision-recall curve (PRC) and area under the receiver operating characteristic (ROC) curve (AUROC), of all methods (Fig. 3). The corresponding PRC and ROC curves (shown in Supplementary Note 2) are estimated on 20 simulated datasets for each setting, estimated by varying the $p$-value threshold, which are obtained by $t$-test in LR, Perm I for NNs, and Perm T for Lasso and XGB.

*The proposed permutation method works well where the existing methods fail for neural networks.* The proposed NN approach has a clearly greater power than the alternative methods (XGB, Lasso, and LR, with both PCA and top-SNP to represent genes), while having roughly the same specificity when interactions are measured with the proposed permutation test (Perm I in Fig. 2a). By contrast, NNs with existing permutation approaches, e.g., Perm R and Perm T in Fig. 2a, have much less specificity in all settings, as the null distributions severely underestimate the interaction scores under the null hypothesis and consider most pairwise interactions positive (Perm R and Perm T in Fig. 2b). Intuitively, interactions are estimated between hidden nodes in the gene layer, and gene representations can change dramatically after excluding main effects which usually represent most of the signal, which makes interaction detection challenging or impossible. We regard interactions as positive if their nominal $p$-value from permutation is less than 0.05 in Fig. 2a (see results with other thresholds in Supplementary Fig. 2). In Fig. 2b, we visualize null distributions of interactions from NNs with different permutation methods for different S/Ns. The null distribution of Perm I can reject false interactions correctly, by retaining the predicted main effect as part of the regression target, while Perm T and Perm R consider both true and false interactions positive.

*NNs can detect complex interactions more accurately than reference methods.* We observe that in all settings the NN has larger AP and AUROC than the baselines in Fig. 3. This is mainly due to the fact that NNs make use of all SNPs to represent genes in a supervised manner (i.e., taking the phenotype into account), which is more informative than selecting a single SNP (e.g., top SNP) or representations learned without supervision (e.g., PCA). Specifically, the second column of Fig. 3 shows that as more individual-level training data are available, NNs can detect interactions more accurately while existing approaches are limited by their strong model assumptions. In the third column, we observe that NNs are close to top-SNP approaches on datasets with small proportions of causal SNPs, where top-SNP methods can approximate the true model well. Although NNs use a sparsity regularization to find the causal SNPs, it requires a large dataset in general. Moreover, as the proportion of causal SNPs increases, NNs and PCA that can use all SNPs become better while top-SNP approaches become worse. In Supplementary Note 2.1–2.5, we provide experimental results when only 2 and 5 genes, instead of all genes, have interaction effects on the phenotype, and we scale up the number of genes to 50 and 100 by modifying the simulator that is defined in Eq. (1). We observe that the NN outperforms existing baselines in these settings as well. In Supplementary Note 2.6, we conduct ablation studies to further explain the success of the proposed method.

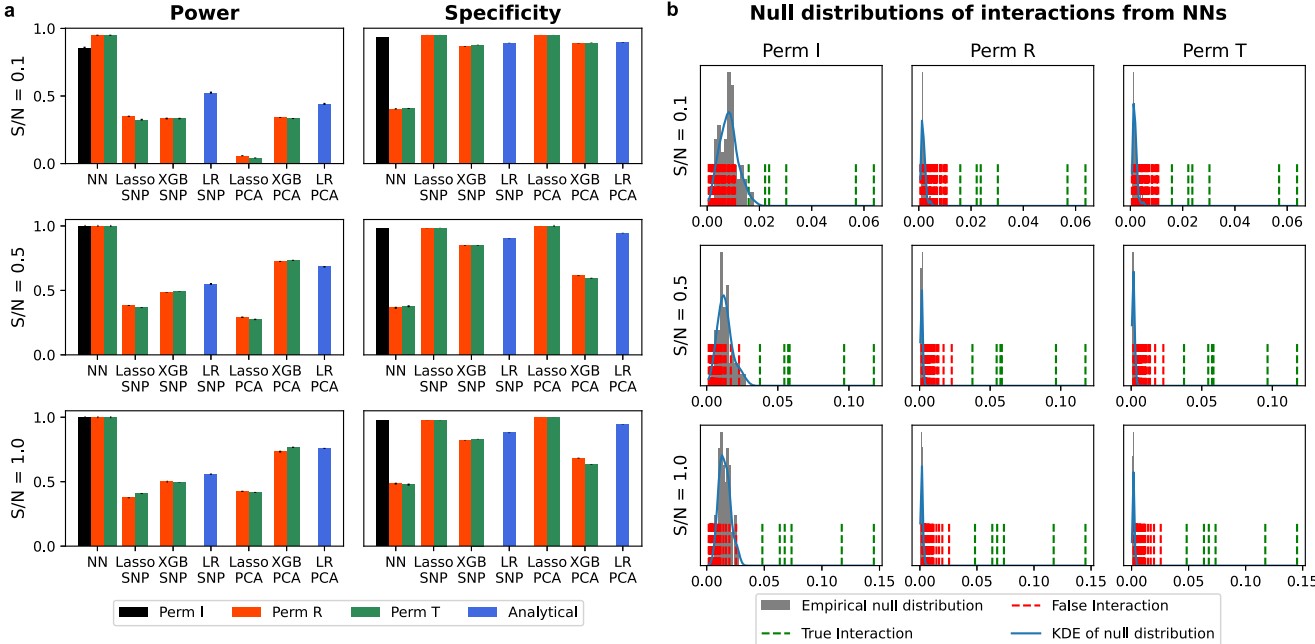

**Fig. 2 Comparison of interaction detection methods with different permutation algorithms. a** Comparison of the power (true positive rate) and specificity (true negative rate) of different methods on simulated datasets with complex interactions for different S/Ns. Error bars are estimated with 20 different simulations. Null distributions of linear regression (LR) are derived analytically (with 80k samples), and null distributions without analytical forms are constructed by three permutation methods (with 50 random permutations) with the maxT correction. We consider interaction as positive (i.e. detected) if its *p*-value obtained from the null distribution is less than 0.05. NNs (black bars) have greater power and similar specificity compared with other approaches when interactions are tested with the proposed Perm I. With existing permutation approaches (Perm R and Perm T), NNs have much smaller specificity in all settings. **b** Examples of null distributions of interactions from NNs for the three different permutation methods on simulated datasets with complex interactions and different S/Ns. We visualize each null distribution with the empirical distribution and the kernel density estimate (KDE). Green/red dashed lines show all true/false interactions. The null distributions generated by Perm I correctly reject false interactions, while Perm T and Perm R consider most false interactions positive.

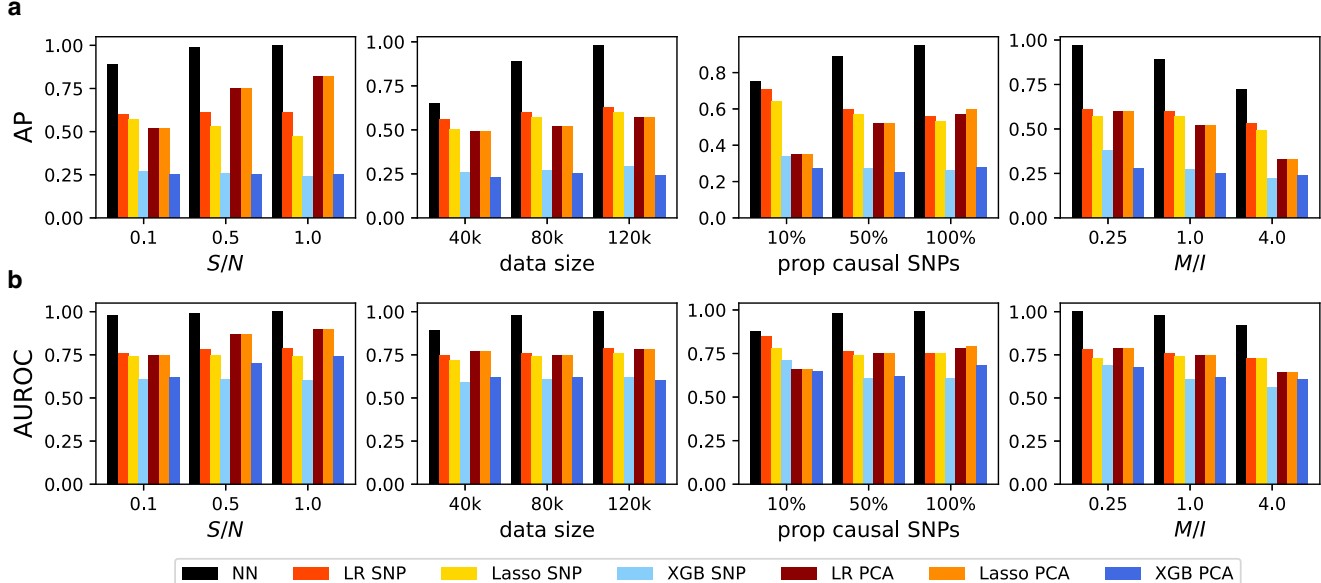

**Fig. 3 Gene–gene interaction detection on simulation datasets with complex interactions.** We show the average precision (AP) and AUROC of each gene–gene interaction detection methods on simulation datasets with different configurations on panel (**a**) and panel (**b**) respectively. We simulate data using S/N = 0.1, datasize = 80*k*, prop causal SNPs = 50%, and M/I = 1.0. We vary one hyper-parameter at a time while fixing others to the specified values. The NN approaches (black bar) are consistently better than existing approaches.

### Simulation with simple interactions

*Setting.* We use the same simulation setting as the simulation with complex interactions except that all interactions are multiplications between the top SNPs. We simulate the phenotype **y** by

first simulating the main effect $\mathbf{w}_i^{\mathrm{main}}$ of each gene *i* as a linear combination of all SNPs in the gene, and then generate multiplicative interactions between the top SNPs (the SNP with the largest main effect) of the genes (Eq. (2)). The simulator is

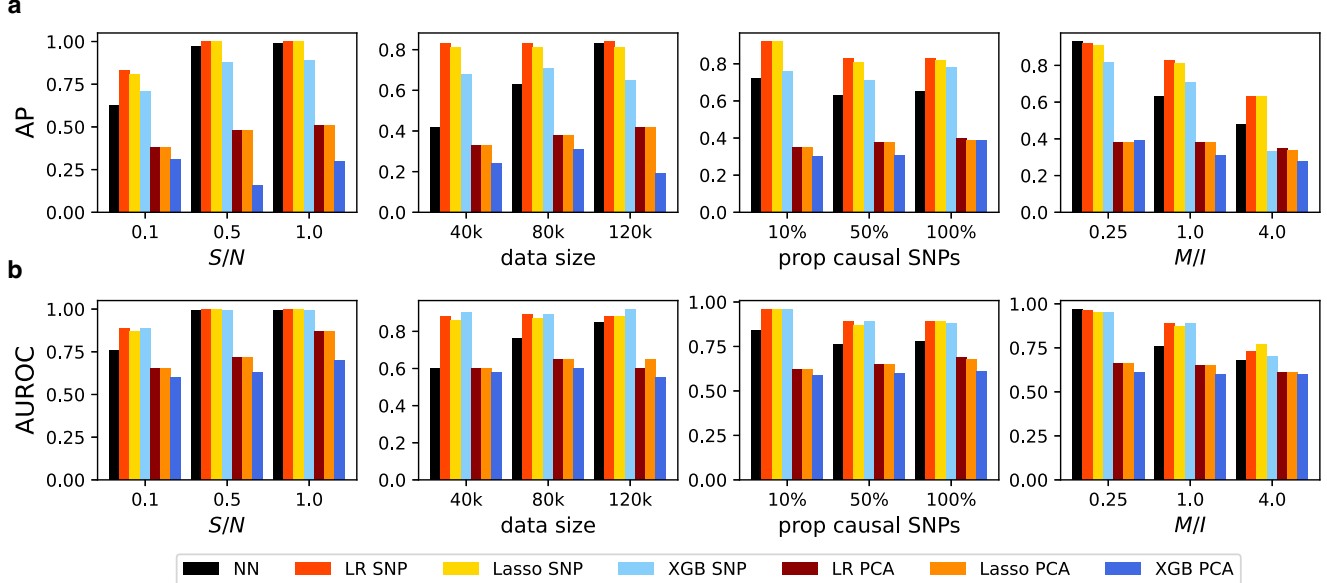

**Fig. 4 Gene–gene interaction detection on simulation datasets with simple interactions.** We show the average precision (AP) and AUROC of each gene–gene interaction detection methods on simulation datasets with different configurations on panel (**a**) and panel (**b**) respectively. We simulate data using S/N = 0.1, datasize = 80$k$, prop causal SNPs = 50%, and M/I = 1.0. We vary one hyper-parameter at a time while fixing others to the specified values (each panel). Methods that use the top-SNP approach to represent genes are usually better than others, including NN.

defined as follows:

$$\mathbf{w}_i^{\text{main}} = \sum_{j=1}^{d_i} \alpha_{ij} \mathbf{x}_{ij}, \quad \forall i \in \{1, \dots, 10\},$$

$$\mathbf{y} = w_{12}\mathbf{x}_{1t_1}\mathbf{x}_{2t_2} + w_{79}\mathbf{x}_{7t_7}\mathbf{x}_{9t_9} + w_{35}\mathbf{x}_{3t_3}\mathbf{x}_{5t_5} + w_{48}\mathbf{x}_{4t_4}\mathbf{x}_{8t_8}$$
$$+ w_{67}\mathbf{x}_{6t_6}\mathbf{x}_{8t_8} + w_{810}\mathbf{x}_{8t_8}\mathbf{x}_{10t_{10}} + \sum_{k}^{10} \mathbf{w}_k^{\text{main}} + \epsilon,$$

$$(2)$$

where $\mathbf{x}_{it_i}$ is the SNP with the largest main effect in gene $i$, i.e., $t_i = \text{argmax}_j |\alpha_{ij}|$.

*Top-SNP approaches have higher power than NNs for simple interactions.* From Fig. 4, we observe that NNs have slightly smaller AP and AUROC compared with the top-SNP approaches to detect simple interactions in general. Because the simulator in Eq. (2) matches the assumption of the simple top-SNP approach, that approach can discover the interacting SNPs precisely. In the third column, we observe that the interaction detection accuracy of any method is not sensitive to the proportion of causal SNPs in the simulator, unlike in the cases with complex interactions in Fig. 3. This is because the proportion of causal SNPs only affects the main effect in Eq. (2), and only one SNP per gene is used to generate interactions. Indeed, as we have already seen in the simulation with complex interactions, NNs outperform all top-SNP approaches consistently when the interactions are complex and the top-SNP assumption does not hold. Moreover, when the data size is large, e.g., 120k, and S/N is high, e.g., 0.5, NNs are comparable with top-SNP approaches even for simple interactions (first two columns in Fig. 4). In Supplementary Note 3, we show the experimental results when the number of interacting genes is small and when the total number of genes is high.

## Simulation without interactions
*Setting.* We construct null simulations, with only linear main effects of genes but without any interactions, to assess the calibration of the proposed permutation procedure. We follow the same setting as previous simulation studies and simulate the

phenotype with

$$\mathbf{y} = \sum_{i}^{10} \sum_{j=1}^{d_i} \alpha_{ij} \mathbf{x}_{ij} + \epsilon. \qquad (3)$$

*The null distribution generated by the permutation is well calibrated.* In the results on Fig. 5, we first observe that the p-values are almost uniformly distributed (inner panels). Moreover, the false positive rate (which is known as there are no interactions, i.e., all tests are false) is consistent with the corresponding p-value in each setting, as shown in the calibration plots (red lines are close to the diagonals), which indicates that the permutation distributions are well-calibrated. Other permutations (Perm R and Perm T) generate poorly calibrated null distributions for NNs, and we show the calibration plots of them in Supplementary Note 4. Moreover, in Supplementary Note 4, we show that the proposed permutation can be combined with the maxT correction to control the false positive rate in multiple testing.

## Interaction discovery in a cholesterol study
*NNs identify 19 candidate gene–gene interactions for three cholesterol phenotypes in the UK Biobank.* We apply the NN interaction detection method on the UK Biobank dataset to demonstrate its ability to identify interactions. We select 424,389 individuals that pass sample quality control with 65 genes which have previously been found to have high main effects on the NMR metabolomics phenotype[26,27]. Note that selecting genes based on their main effects does not bias the assessment of the significance of interactions using permutation. As shown in the simulation study without interactions, the permutation of interactions is well-calibrated when the main effects exist. We select all SNPs (17,168 SNPs, after quality control) included within the 65 chosen genes with 10kbp flanking regions added to both ends of each gene, and we select HDL (high-density lipoprotein), LDL (low-density lipoprotein), and TC (total cholesterol) as dependent variables. To preprocess each phenotype, we first regress out age and gender, and then we regress out the first 10 genotypic principal components to correct for population structure. We provide further preprocessing details in the Methods section.

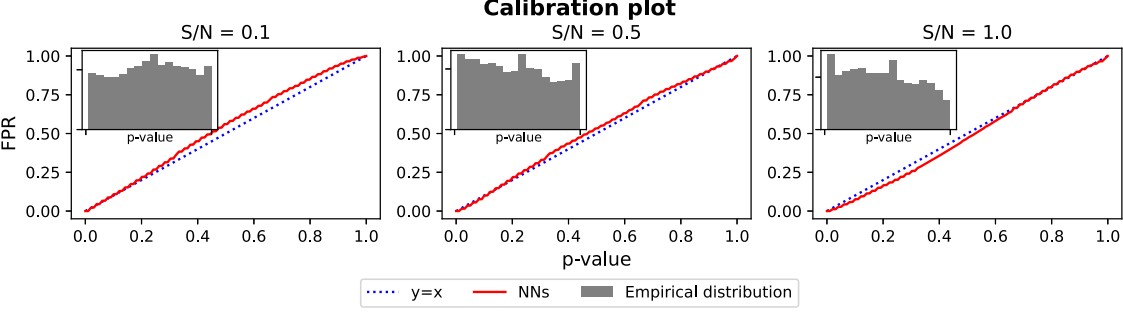

**Fig. 5 Calibration plots of the proposed permutation for NNs on simulation datasets without interactions.** Inner panels show that empirical distributions of *p*-values (with 50 random permutations) are uniform, and red lines show that the *p*-value from permutation is close to the corresponding false positive rate (blue dashed lines). This means the proposed permutation distribution for NNs interactions is well-calibrated.

After we train the model on the training set, gene–gene interaction scores are estimated on the test set. We rank the interaction pairs according to their interaction scores from high to low, and we construct null distributions via the permutation test. We show the histograms of false discovery rates (FDR) of interactions in Fig. 6a. By setting the FDR threshold to 0.1, we discover 19 candidate interactions for three cholesterol phenotypes in total, which means that 17 out of 19 candidate interactions are expected to be true findings. The Q–Q plots of interaction scores on observed versus permuted datasets are shown in Fig. 6b, from which we observe that the candidate interactions (marked with blue) of each phenotype significantly deviate from the expectation under the null hypothesis. We list all candidate interactions of each phenotype and their corresponding interaction scores in the first two columns of Table 1.

Next we investigate if the candidate interactions detected by the NN could have been found by other methods. For the earlier method referred to as the top-SNP method, we train a bivariate LR with a multiplicative interaction term for each candidate,

$$\hat{\mathbf{y}}^{B} = \hat{\beta}_0^{B} + \hat{\beta}_i^{B} \tilde{\mathbf{g}}_i^{B} + \hat{\beta}_j^{B} \tilde{\mathbf{g}}_j^{B} + \hat{\beta}_{i,j}^{B} \tilde{\mathbf{g}}_i^{B} \tilde{\mathbf{g}}_j^{B}, \tag{4}$$

where the $\tilde{\mathbf{g}}_i^{B}$ and $\tilde{\mathbf{g}}_j^{B}$ denote genes *i* and *j* represented by their top SNPs, and the superscript B refers to the UK Biobank dataset (in contrast with superscript F that will be later used for the FINRISK data). The significance of coefficient $\hat{\beta}_{i,j}^{B}$ is used to check if the interaction could be detected by the top-SNP method, and the *p*-values are reported in the 4th column of Table 1. We see that only 6 out of 19 candidate interactions were nominally significant using the top-SNP approach. This indicates that NNs can find interactions missed by the standard approach.

We further check if any interactions have already been observed in previous studies, shown in the last column of Table 1). For example, no interaction effect between *LIPC-CETP* and *ABCA1-CETP* were found in a Japanese[28] and a Chinese Han cohort[29] respectively, where only one SNP was used to represent the gene in studies. Some of the interactions should exist according to the underlying biological mechanisms, such as *LIPC-CETP*[30] and *LDLR-APOE*[31], but they were not found in any association study. Moreover, we find interactions already existing in KEGG[32] cholesterol metabolism pathway, such as *LPL-CETP* and *ABCA1-CETP*. Finally, we observe interactions detected on LDL are similar with interactions detected on TC, because LDL usually makes up 70% of the TC in human body.

*Nine of the candidate interactions replicate in the FINRISK dataset.* We use an independent dataset, FINRISK, to verify if the 19 candidate interactions detected in the UK Biobank are replicable (see Methods for data preprocessing). We select the same SNPs from the FINRISK, and calculate gene representations

according to the model learned with the UK Biobank. For replication, we first calculate the residual in the FINRISK by subtracting the main effects from the phenotype value for a given pair of genes, i.e., $\mathbf{y}^{F} - (\hat{\beta}_0^{F} + \hat{\beta}_i^{F} \mathbf{g}_i^{F} + \hat{\beta}_j^{F} \mathbf{g}_j^{F})$. Then, we estimate the interaction pattern for the gene pair in the UK Biobank by training two NNs: a fully connected MLP, $\hat{\mathbf{y}}^{B} = h^{B}(\mathbf{g}_i^{B}, \mathbf{g}_j^{B})$; and another NN that only contains non-linear main effects without any interactions, $\hat{\mathbf{y}}^{B} = h_i^{B}(\mathbf{g}_i^{B}) + h_j^{B}(\mathbf{g}_j^{B})$. We represent the interaction learned from the UK Biobank by the difference between these two NNs, $h^{B}(\cdot, \cdot) - h_i^{B}(\cdot) - h_j^{B}(\cdot)$, i.e., this is the part that cannot be explained by main effects. Finally, we use the following linear regression to test if the interaction pattern learned from the UK Biobank can explain the observed residual in the FINRISK:

$$
\begin{aligned}
&\underbrace{\mathbf{y}^{F} - \left(\hat{\beta}_0^{F} + \hat{\beta}_i^{F} \mathbf{g}_i^{F} + \hat{\beta}_j^{F} \mathbf{g}_j^{F}\right)}_{\text{Observed residual in FINRISK}} \\
&= \alpha_0^{F} + \alpha_{i,j}^{F} \underbrace{\left(h^{B}\left(\mathbf{g}_i^{F}, \mathbf{g}_j^{F}\right) - h_i^{B}\left(\mathbf{g}_i^{F}\right) - h_j^{B}\left(\mathbf{g}_j^{F}\right)\right)}_{\text{Predicted interaction in FINRISK}} + \epsilon_{i,j}.
\end{aligned}
\tag{5}
$$

If $\alpha_{i,j}^{F}$ is significantly greater than zero (with one-side *t*-test), we regard the interaction between genes *i* and *j* as replicated. We consider the above tests for replication because the FINRISK dataset is too small (4620 individuals in total) to train any neural network well with the same number of features as the UK Biobank. The replication *p*-values for each interaction are shown in Table 1. Out of the 19 possible interactions, nine finally replicate significantly in the FINRISK shown in the 5th column of Table 1, with a FDR around 0.022. We further check if the replicable interactions could be learned by the top-SNP method, Eq. (4), on the FINRISK dataset directly. The *p*-values are reported in the 6th column of Table 1 and they verify that only one of these interactions is found significant with the top-SNP method.

In Fig. 7, we visualize one replicated interaction per phenotype and its corresponding fit for the linear regression in Eq. (5) (see the remaining replicable interactions in Supplementary Note 5). From the linear regression (Fig. 7a), we notice that the interaction pattern learned from the UK Biobank can explain the observed residual in the FINRISK. We use blue, purple, and yellow background colorings to represent individuals with low, medium, and high interaction values respectively, and mark those individuals in the visualization of the learned interaction pattern with the same color. From the visualizations (Fig. 7b), we notice that the shape of the low interaction regions, the pink, can be complex, which may not be approximated well by a multiplicative interaction. For example, the interaction effect between *LIPC* and *LIPG* on HDL is positive only if the representation of *LIPC* and

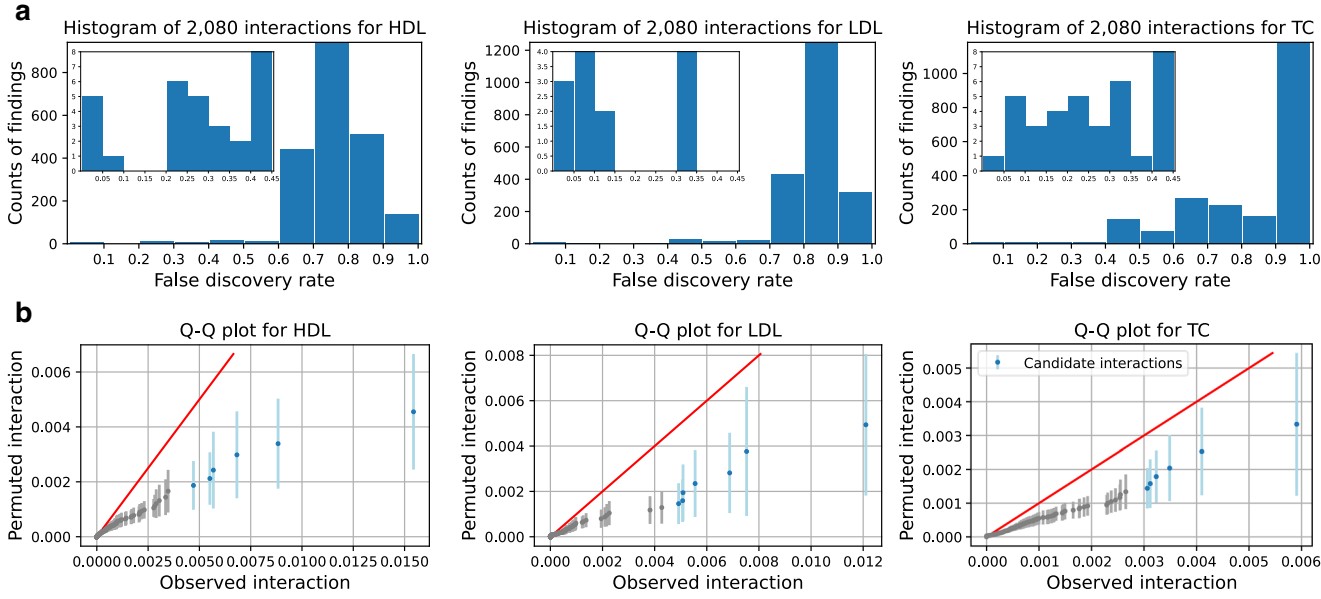

**Fig. 6 Statistics of gene–gene interactions detected by NNs in a cholesterol study with the UK Biobank dataset. a** The histogram of interactions for different false discovery rates (FDR). The inner panel zooms into FDR < 0.45 region. Each bar represents how many more interactions we will consider as positive if we increase FDR; e.g., 6 more findings are included by increasing FDR from 0.20 to 0.25 for HDL. **b** The Q-Q plot of Shapley interaction scores on observed vs. permuted data. The bars cover 95% of interaction scores under the null hypothesis. We observe that top candidate interactions (blue, with FDR < 0.1) clearly deviate from the null distribution.

**Table 1 Candidate interactions detected by NNs in the real-world cholesterol study.**

| Phenotype | Genes | Interaction Score (FDR) | Multiplicative top SNPs, *p*-value | FINRISK, NN replication, *p*-value | FINRISK, Multiplicative top SNPs, *p*-value | Reference |
|---|---|---|---|---|---|---|
| HDL | *LPL, CETP* | 1.543 (0.000) | 0.100 | 0.070 | 0.895 | Yes[32,42] |
| | *LIPC, CETP* | 0.883 (0.000) | **0.034** | **0.009** | 0.532 | No[29], Yes[30] |
| | *CETP, LIPG* | 0.682 (0.011) | 0.424 | 0.095 | **0.030** | Yes[43] |
| | *ABCA1, CETP* | 0.568 (0.050) | 0.207 | 0.138 | 0.954 | No[28] |
| | *LPL, LIPC* | 0.551 (0.046) | 0.434 | 0.145 | 0.180 | NA |
| | *LIPC, LIPG* | 0.471 (0.099) | 0.474 | **0.000** | 0.176 | NA |
| LDL | *LDLR, APOE* | 1.210 (0.000) | **0.020** | **0.022** | 0.841 | Yes[31] |
| | *SORT1, APOE* | 0.752 (0.047) | 0.159 | **0.000** | 0.174 | NA |
| | *LPA, APOE* | 0.688 (0.049) | **0.020** | 0.057 | 0.889 | Yes[44] |
| | *TRIB1, APOE* | 0.555 (0.060) | **0.008** | **0.000** | 0.485 | NA |
| | *HMGCR, APOE* | 0.509 (0.062) | 0.113 | **0.000** | 0.395 | NA |
| | *APOB, APOE* | 0.507 (0.071) | 0.081 | 0.072 | 0.970 | Yes[45] |
| | *PCSK9, APOE* | 0.491 (0.097) | 0.402 | 0.152 | 0.394 | NA |
| TC | *LDLR, APOE* | 0.591 (0.034) | **0.039** | **0.001** | 0.750 | Yes[31] |
| | *PCSK9, APOE* | 0.410 (0.090) | 0.820 | **0.000** | 0.188 | NA |
| | *SORT1, APOE* | 0.349 (0.096) | 0.222 | 0.099 | 0.124 | NA |
| | *LIPG, APOE* | 0.323 (0.082) | 0.200 | 0.089 | 0.600 | NA |
| | *LIPC, APOE* | 0.311 (0.076) | 0.146 | **0.000** | 0.472 | Yes[46] |
| | *APOE, APOC1* | 0.306 (0.066) | **0.001** | 0.513 | 0.351 | NA |

First column shows the three phenotype in the study. Second column shows the corresponding pair of genes, third column the interaction score and the false discovery rate. The *p*-values using the top-SNP method are shown in the fourth column, where bold highlights *p*-value <0.05 (with 424,389 independent samples). The replication results on FINRISK (*p*-values, with 4620 independent samples) are shown in the fifth column. The *p*-values (with 4620 independent samples) using the top-SNP method on FINRISK are shown in the sixth column, and only one interaction is significant. The last column gives a reference from previous literature (if any), where 'Yes' and 'No' denote whether the interaction is known/found or not, and 'NA' denotes that there are no related references.

*LIPG* are both high, i.e., an 'AND' interaction between *LIPC* and *LIPG*. Moreover, the interaction between *PCSK9* and *APOE* on TC has a complex multi-modal shape. We further notice that even if these interactions are replicated significantly, the effect sizes are relatively small, increasing $R^2$ on average by 1.3% (ranging from 0.2% to 3.9%) on the test set. However, compared with the existing top-SNP linear regression, NNs increase the test $R^2$ significantly, on average by 36.7%, ranging from 4.0% to 109.8% (see details in Supplementary Note 6).

## Discussion

Existing approaches for detecting interactions in a GWAS do not reveal all interactions between a given set of genes because 1. genes are typically represented by the single most significant SNP, and 2. only restricted forms of interaction (e.g. multiplicative) are considered. Therefore, the statistical power can be limited for complex gene–gene interactions and important interactions may be ignored. Here, we resolved these shortcomings with a deep learning approach. NNs are generally well-known for their ability

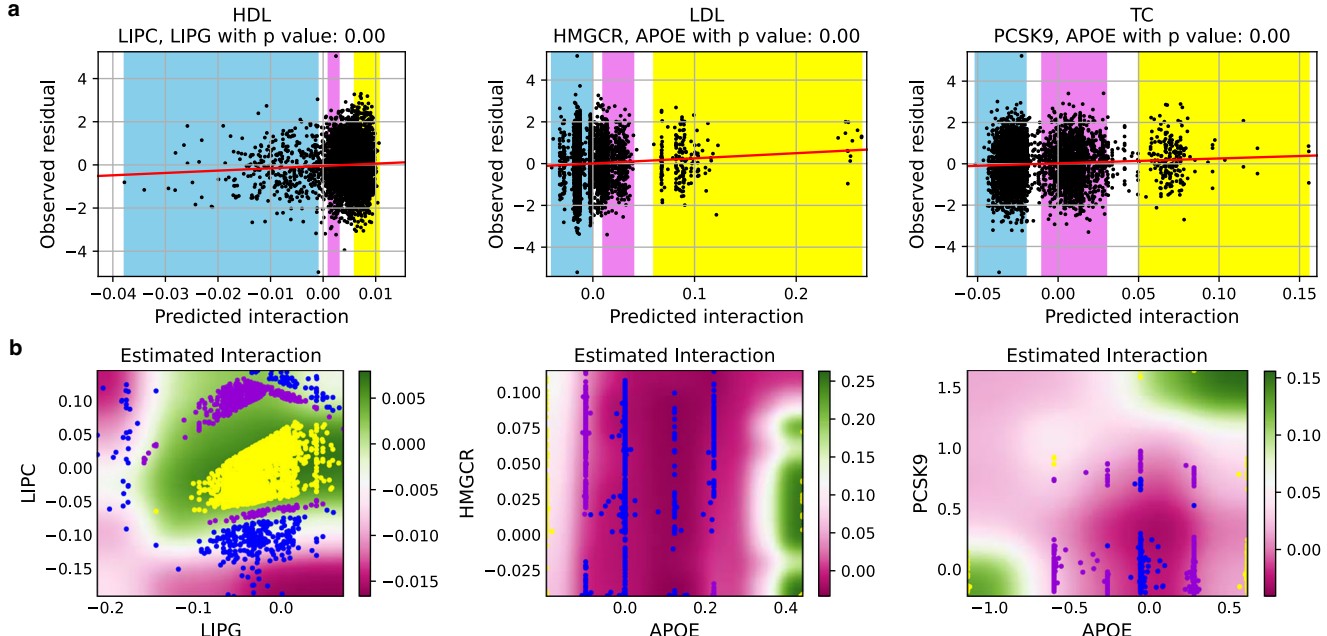

**Fig. 7 Visualization of one interaction replicated in the FINRISK dataset of each phenotype. a** X axis represents the interaction predicted in FINRISK using the model trained on the UK Biobank. Y-axis shows the residual after removing the main effects in the FINRISK dataset; see Eq. (5). The interactions learned from the UK Biobank (x-axis) significantly explain the variation in the phenotype of FINRISK that remains after excluding the main effects (y axis). **b** Heatmaps representing the estimated interaction (background color) between two gene representations (x-axis and y-axis) learned from the UK Biobank. Blue, purple, and yellow backgrounds in the panel (**a**) represent individuals with low, medium, and high predicted interaction values, and they are shown with dots in the panel (**b**). It is clear that the replicable interactions have complex shapes which can not be approximated well by multiplicative interactions.

to learn high level representations from low level features with suitable architectures that reflect prior knowledge of the application domain. Furthermore, as a universal function approximator, a NN can model any flexible relation between genes and phenotypes, including interactions. Despite these benefits, NNs have not been widely used to detect interactions in GWAS because it is neither trivial to uncover the interactions from a NN nor assess the significance of the findings. We defined a gene interaction NN using a structured sparse architecture, which learned a representation of each gene from all the corresponding SNPs, instead of considering a designated top SNP or ignoring information in the phenotype (as, e.g., PCA, does). From this network, by applying a mathematically principled interpretation method, Shapley interaction scores, we were able to estimate the flexible gene–gene interactions as a post-hoc step. In addition, we designed a permutation procedure to assess the statistical significance of interactions, which is essential to control the false discovery rate of findings.

The proposed deep learning framework has many attractive properties, but also some shortcomings. First, the framework could easily be extended to detect higher-order interactions by calculating higher-order Shapley interaction scores, but this would require $O(M^t)$ forward passes for $t$-order interactions among $M$ genes, and therefore computation could be a bottleneck. Second, it is computationally infeasible to detect gene–gene interactions in the whole-genome wide scale with common architectures due to the extremely high dimension of the genome. We resolved this problem by detecting interactions between a set of candidate genes which had previously been found to have large main effects, but such a simplification can ignore interactions between genes that were not selected. A better solution could be to devise a tailored neural network architecture for the whole genome SNP data or accommodating interaction detection into distributed deep learning frameworks. Third, we noticed that

although NNs can detect interactions that current methods ignore, the estimated effect sizes are relatively small. This could be improved by improving the architecture of the NN or using more informative priors, such as modeling known interactions from existing genetic networks[33,34] with graph neural networks, but how to learn new interactions from data with predefined graphs is still an interesting open question. Lastly, we point out that although NNs have the highest power among various baselines on simulated datasets with complex interactions, the power of NNs is smaller than pairwise top-SNP approaches on the UK Biobank in general (Supplementary Fig. 24). We hypothesise that most interactions in the real-world data can be approximated well by multiplication between top SNPs, i.e., simple interactions, and top-SNP methods have higher power than NNs. Therefore NN should not used as a substitute for the default top-SNP approach, but rather as a complementary tool to find interactions that would otherwise be missed. Indeed, with real world genetic data, we showed that our NN Framework can find interactions that cannot be detected via ordinary SNP based approaches, for example, 5 of the 9 replicable interactions could not be detected by a linear regression with top-SNP representations (Fig. 7).

## Methods

**Data preprocessing**. UK Biobank (UKB) is a prospective cohort of approximately 500,000 individuals from the United Kingdom, which consists of rich genotype and phenotype information of each individual[15]. We select 424,389 individuals that pass sample quality control with 65 genes which have previously been found to have high main effects on cholesterol phenotypes such as the HDL, LDL, and TC[26,27], and 10 out of 65 genes are used on simulated datasets. We select all SNPs included within the 65 chosen genes with 10kbp flanking regions added to both ends of each gene. Quality control is conducted by filtering out SNPs with missing rate > 0.05, MAF < 0.01, pHWE < $10^{-6}$ and info score (imputation quality) < 0.4, leaving 17,168 SNPs. We use HDL, LDL, and TC as the target phenotypes to be associated with the selected genes. To reduce collinearity, we prune SNPs to keep 90% of total variance according to the linkage disequilibrium (LD) score[35], finally amounting to 4322, 3776, and 4264 SNPs for HDL, LDL, and TC respectively.

We regress out age and gender covariates as well as the first 10 genotypic principal components to remove population structure. The residuals are quantile normalized before using them as the dependent variable, and the SNPs are centered before using them as independent variables. On both simulated and real data, we split the preprocessed data into 70% for training and 30% for testing.

To replicate the positive gene–gene interactions detected from the UK Biobank, we select an independent dataset from the FINRISK project (DILGOM07 subset) which includes a cohort of unrelated individuals aged 30-59 years participating in a study on coronary heart diseases in Finland[16]. We select 4,620 individuals that pass the sample quality control, and select the same set of SNPs as in UK Biobank for each gene in order to reuse the NN gene representations learned on the UK Biobank dataset. We choose HDL, LDL, and TC as regression targets, and apply the same preprocessing steps on phenotypes as we did on the UK Biobank.

## Interaction detection with deep learning

*Notation.* We assume that there are $N$ individuals and $M$ genes with $m_i$ SNPs for gene $i$, and $P = \sum_{i=1}^{M} m_i$ SNPs in total. We denote the target variable, i.e., one chosen phenotype, by a vector $\mathbf{y} = (y_1, \ldots, y_N)^T$ and the covariate matrix containing all SNPs of the selected genes by $\mathbb{X} = (\mathbf{X}_1, \ldots, \mathbf{X}_M)$, where $\mathbf{X}_i = (\mathbf{x}_{i1}, \ldots, \mathbf{x}_{im_i})$ represents the matrix of SNPs of gene $i$ and $\mathbf{x}_{ij} = (x_{ij1}, \ldots, x_{ijN})^T$ is an $N$-dimensional column vector that represents the $j$th SNP of gene $i$ for all individuals. We use $X_k$ to represent a $P$-dimensional row vector of all SNPs of individual $k$, such that $\mathbb{X} = (X_1^T, \ldots, X_N^T)^T$.

*Gene interaction NNs.* The architecture of a NN reflects inductive biases of the model, which is one of the keys to the success of NNs. In genetics, the effect of SNPs on a phenotype can be abstracted into a two-stage procedure: 1. SNPs within a gene affect how the gene behaves; 2. The combined behaviors of multiple genes affects the phenotype eventually. We propose a NN architecture with structured sparsity that leverages these two mechanisms. Mathematically, a structured sparse neural network model can be written as

$$\mathbf{g}_i = h_{g_i}(\mathbf{X}_i; \mathbf{w}_{g_i}), \quad \forall i \in \{1, \ldots, M\}; \tag{6a}$$

$$\mathbf{y} = h_p(\mathbf{g}_1, \ldots, \mathbf{g}_M; \mathbf{w}_p) + \epsilon, \tag{6b}$$

where $h_{g_i}(\cdot)$ represents a multilayer perceptron (MLP) parametrized by $\mathbf{w}_{g_i}$ and the output $\mathbf{g}_i$ is an $N$-dimensional vector that learns the representation of gene $i$ from its SNPs $\mathbf{X}_i$. $h_p(\cdot)$ is another MLP that models the relations between genes and the phenotype $\mathbf{y}$, and $\epsilon$ represents the part of the phenotype that cannot be explained by genetic information. We demonstrate the architecture in Fig. 1, where the hidden layer that contains gene representations $\mathbf{g}$ is called the gene layer. The architecture is structured sparse because SNPs from different genes are disconnected until the gene layer. All MLPs in Eq. 6 are trained in an end-to-end manner by reformulating the model as

$$\mathbf{y} = h_p(h_{g_1}(\mathbf{X}_1), \ldots, h_{g_M}(\mathbf{X}_M)) + \epsilon = f(\mathbb{X}; \mathbf{w}) + \epsilon, \tag{7}$$

where $\mathbf{w}$ represents all parameters in the model. Eq. (7) only takes the SNPs $\mathbb{X}$ and phenotype $\mathbf{y}$ as inputs during training. Empirically, we find that using small MLPs for both $h_{g_i}(\cdot)$ and $h_p(\cdot)$ usually has a better accuracy than large architectures.

We use 3 hidden layers in the model: one hidden layer MLP with architecture $m_i - 10 - 1$ (i.e., $m_i$ nodes in the input layer, 10 nodes in the hidden layer, and 1 node in the output layer) for $h_{g_i}(\cdot)$ and $10 - 100 - 1$ for $h_p(\cdot)$ in simulation datasets, and $m_i - 10 - 1$ for $h_{g_i}(\cdot)$ and $65 - 200 - 1$ for $h_p(\cdot)$ in the UK Biobank dataset, where 10 and 65 are the number of genes considered in each experiment. We use ReLU as the activation function. We regularize weights in $h_{g_i}(\cdot)$ and $h_p(\cdot)$ with $L_1$ regularization, because the proportions of causal SNPs and genes might be small. We use the mean squared error (MSE) between predicted phenotype and observed phenotype as the loss function, and we cross-validate the regularization strength on the training set with 5-fold cross-validation. We use early-stopping to determine the number of epochs and we set the batch size to 30,000. We train models with Adam with learning rate 0.005.

*Deep ensemble of NNs.* The loss surface of NNs is highly multi-modal, and the stochastic optimization methods used to learn the posterior can only find a single mode[17]. An easy way to capture multiple modes and thus improve the model accuracy is through a deep ensemble[36], which repeats the same training procedure multiple times with different initialization of parameters and then averages different NNs during testing. We apply this trick for all experiments, where we use an ensemble of 50 NNs trained with different initialization. Deep ensemble is computationally inefficient, but it can be parallelized easily.

*Gene–gene interaction score.* Shapley interaction score[10] is a well-axiomatized interaction score for input features motivated by the Shapley values in game theory. We denote the set of all features by $F$, a feature $i \in F$, and a feature set $S \subseteq F$. We define the interaction effect between feature $i$ and $j$, with feature set $S$, of a neural

network $f$ at a data point $X_k$ to be

$$\delta_{ij}^f(X_k; S) = f(X_k; S \cup \{i,j\}) - f(X_k; S \cup \{i\}) - f(X_k; S \cup \{j\}) + f(X_k; S), \tag{8}$$

where $f(X_k; S)$ is the prediction at $X_k$ when only features in $S$ are used, which often requires retraining the NN multiple times. A common approximation is to replace the absent features (i.e., $F \backslash S$) by the corresponding values in a baseline $C_{F \backslash S}$, such that

$$f(X_k; S) \approx f(X_{K,S}; C_{F \backslash S}). \tag{9}$$

The baseline is usually set as the empirical mean of each feature, which is used in this work as well, but more informative baselines can also be applied. The Shapley interaction score $\mathrm{SI}_{ij}^f(X_k)$ is the expectation of $\delta_{ij}^f(X_k; S)$,

$$\mathrm{SI}_{ij}^f(X_k) = \mathrm{E}_{p(S)}[\delta_{ij}^f(X_k; S)], \tag{10}$$

over a uniformly random chosen feature set $S$ from $F$. This could be computationally expensive when the dimension $|F|$ is high. We use a Monte-Carlo procedure[37] to approximate $\mathrm{SI}_{ij}^f(X_k)$ by a small number of samples of $S$.

Shapley interaction score $\mathrm{SI}_{ij}^f(X_k)$ is a *local* interaction effect measure, representing the interaction effect between feature $i$ and $j$ of function $f(\cdot)$ at data point $X_k$. To aggregate the *local* interaction effect at different data points into a *global* interaction effect, i.e., shared by the whole data domain, we use take the expectation of $|\mathrm{SI}_{ij}^f(X_k)|$ w.r.t. the empirical data distribution $p(X)$, such that

$$\mathrm{SI}_{ij}^f = \mathrm{E}_{p(X)}[|\mathrm{SI}_{ij}^f(X)|]. \tag{11}$$

Instead of the interaction between individual input features (SNPs), we are interested in interactions between genes, which are represented by the hidden nodes of the gene layer. Thus the interaction between genes $i$ and $j$ can be calculated by the Shapley interaction score between between $\mathbf{g}_i$ and $\mathbf{g}_j$ in $h_p(\mathbf{g}_1, \ldots, \mathbf{g}_M; \mathbf{w}_g)$ on Eq. (6b), where the gene representations, $\mathbf{g}$, are calculated in lower layers on Eq. (6a). For a deep ensemble of NNs, instead of having a point estimator of function $f(\cdot; \mathbf{w})$, we have a posterior distribution of functions $q(f)$ induced by the ensemble distribution of the weight $q(\mathbf{w})$. Thus the interaction score of NNs is the expectation of $\mathrm{SI}_{ij}^f$ w.r.t. $q(f)$, which we estimate by taking the average of $N_f$ samples drawn from the ensemble:

$$\mathrm{SI}_{ij} = \mathrm{E}_{q(f)}[\mathrm{SI}_{ij}^f] \approx \frac{1}{N_f} \sum_{k=1}^{N_f} \mathrm{SI}_{ij}^{f_k}. \tag{12}$$

In experiments, we have in total 50 NNs in the deep ensemble to estimate the interaction scores between genes.

*Permutation test of interaction.* The alternative hypothesis that we want to verify is that interaction effect between genes are non-zero. Thus the corresponding null hypothesis is: genes only have main effects on the phenotype without any interactions. We use a structured sparse linear regression (shown in Fig. 1) to represent the null hypothesis model (i.e., main effects of genes):

$$\mathbf{g}_i = h_{g_i}(\mathbf{X}_i; \mathbf{w}_{g_i}), \quad \forall i \in \{1, \ldots, M\}; \tag{13a}$$

$$\mathbf{y} = w_0 + \sum_{i=1}^{M} w_i \mathbf{g}_i + \epsilon, \tag{13b}$$

where the NN $h_p(\cdot)$ in Eq. (6b) is replaced by a linear regression. When $h_{g_i}(\cdot)$ is reduced to a linear regression, the null hypothesis (Eq. (13b)) will be a reduced rank regression[26] with rank $M$. We propose a permutation method to generate permutation datasets for NN training. We first permute the residual of the null hypothesis (Eq. (13b)), which removes any interaction effect from the data. Then we define the dependent variable as the sum of the predicted phenotype of Eq. (13b) and the permuted residual, which retains the main effect. We do not permute the independent variable. The permutation procedure (**Perm I**) can be summarized in Fig. 8a. We compare against two existing permutation methods, Perm T (Fig. 8b) and Perm R (Fig. 8c), on simulated data. Compared with Perm I, Perm T removes the interaction and the main effect; Although Perm R only permutes the interaction effect, it fails to keep the main effects intact in the permuted datasets. Note that although the Bayesian approach[9,38,39] could be used to quantify uncertainty and assess significance in principle, the posterior distributions of Bayesian NNs obtained by scalable inference algorithms, such as variational inference and stochastic HMC, are poorly calibrated[40]. Moreover, the use of posterior distributions to address the multiple testing problem is still an active research topic. Hence, we use permutation testing to assess significance which is a well-established method in the context of GWAS.

In the permutation procedure, we incorporate two approaches for multiple testing correction : Max-T method[23] and false discovery rate (FDR) control[24], in simulated datasets and real-world applications, respectively. For Max-T method, we collect the maximal Shapley interaction score across all interaction scores for each permutation in Step 4. This provides a single empirical null distribution for all interaction pairs to conduct hypothesis tests. For FDR, we record the Shapley interaction scores of all interaction pairs from high to low for each permutation in Step 4, which allows us to calculate the FDR given a decision threshold of

**a**

| Perm I: Proposed permutation test for interactions |
| --- |

1: Estimate residuals, $\mathbf{r} = \mathbf{y} - \hat{\mathbf{y}}$, by fitting the null hypothesis model in Equation 15;
2: Permute the interaction effect of data by permuting $\mathbf{r}$ to obtain $\mathbf{r}^{\pi}$;
3: Obtain the permutation target by $\mathbf{y}^{\pi} = \hat{\mathbf{y}} + \mathbf{r}^{\pi}$;
4: Compute Shapley interaction score of NNs (Equation 6) trained on permutation datasets $\{(\mathbb{X}, \mathbf{y}^{\pi})\}$;
5: Repeat Steps 2-4 several times to obtain an empirical null distribution.

**b**

| Perm T: use permuted dependent variable as target |
| --- |

1: Permute the dependent variable $\mathbf{y}$ to obtain $\mathbf{y}^{\pi}$
2: Compute Shapley interaction score of NNs (Eq.6) trained on permutation datasets $\{(\mathbb{X}, \mathbf{y}^{\pi})\}$;
3: Repeat Steps 1-2 several times to obtain an empirical null distribution.

**c**

| Perm R: use permuted residual as target |
| --- |

1: Estimate residuals, $\mathbf{r} = \mathbf{y} - \hat{\mathbf{y}}$, by fitting the null hypothesis model in Eq.15;
2: Permute the interaction effect of data by permuting $\mathbf{r}$ to obtain $\mathbf{r}^{\pi}$;
3: Obtain the permutation target by $\mathbf{y}^{\pi} = \mathbf{r}^{\pi}$;
4: Compute Shapley interaction score of NNs (Eq.6) trained on permutation datasets $\{(\mathbb{X}, \mathbf{y}^{\pi})\}$;
5: Repeat Steps 2-4 several times to obtain an empirical null distribution.

**Fig. 8 Permutation algorithms. a** The proposed permutation algorithm for interactions (Perm I). **b** Use permuted dependent variable as target (Perm T). **c** Use permuted residual as target (Perm R).

interaction score. FDR has less stringent control of type I error but greater statistical power than the Max-T method. Permutation testing is computationally heavy as it requires training multiple NNs. Fortunately, like a deep ensemble, permutations are easy to parallelize by applying different random seeds to generate local permutation targets.

**Statistics and reproducibility**. We conducted statistical analysis mainly with Python (https://www.python.org/) software. Additional software including QCTOOL (https://www.well.ox.ac.uk/gav/qctool/) and Plink (https://www.cog-genomics.org/plink/). The specific statistical hypothesis testing methods (one- or two-side $t$-tests) are described in the legends of the corresponding figures. Adjustment for multiple testing was carried out using Max-T correction and False Discovery Rate (FDR) control. $P$-value $< 0.05$ and FDR $< 0.1$ were considered statistically significant for these comparisons.

**Reporting summary**. Further information on research design is available in the Nature Portfolio Reporting Summary linked to this article.

## Data availability
UK Biobank[15] data are available to registered investigators under approved applications [http://www.ukbiobank.ac.uk]. A permission to use the FINRISK[16] data can be applied from THL (Finnish Institute for Health and Welfare). Other relevant data are available from the corresponding author upon request. The source data behind the graphs in the paper are available in Supplementary Data 1.

## Code availability
The source code[41] is available at [https://github.com/tycui/GWAS_NN] under the MIT License.

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

## Acknowledgements

The work used computer resources of the Aalto University School of Science Science-IT project. This work was supported by the Academy of Finland (Flagship programme: Finnish Center for Artificial Intelligence, FCAI, and grants 319264, 292334, 286607, 294015, 336033, 321356), the EU Horizon 2020 (grant no. 101016775), and UKRI Turing AI World-Leading Researcher Fellowship, EP/W002973/1. This research has been conducted using the UK Biobank Resource under application number 46791.

## Author contributions

P.M. and S.K. designed the study. T.C., K.M., J.R., P.M. and S.K. developed the method. T.C. implemented the method and ran all the experiments. K.M. and A.H. preprocessed the datasets. A.H., P.M. and S.K. supervised the work. T.C. wrote the article with contributions from all authors.

## Competing interests

The authors declare no competing interests.
