## [Peer Review File · Communications Biology]

Reviewers' comments:

Reviewer #1 (Remarks to the Author):

The paper entitled "Gene-Gene Interaction Detection with Deep Learning" presents a Deep Learning (DL) approach to identify interactions between pairs of genes given a phenotype and a set of genes related to the phenotype under study. In particular, they design a Neural Network architecture presenting structured sparsity that allows to learn gene representations from the SNPs and, in a deeper layer, learns complex interactions, quantified by calculating the Shapley interaction score, between the genes and the phenotype. Moreover, the authors propose a permutation technique to assess the statistical significance of the detected interactions.

Overall impression of the work

The paper is overall very well written and well structured. The method is interesting and timely as there is growing interest in expanding GWAS tools to be more comprehensive and capable of detecting interactions at SNP level and/or gene level. Moreover, learning gene representations from the SNPs is a very interesting and valuable aspect of the work, since the gene representations that are available/used nowadays in the literature often seem to be overly simplistic (e.g., top associated SNP to represent a gene). However, there are few aspects of the method and its objectives that should be clarified; more details in the subsequent lines. The simulations and analysis on natural phenotype are quite comprehensive, some suggestions for improvement in the following. Furthermore, some advice for improvement concerning the evaluation framework are reported as well.

Specific comments:

1) Both the simulations and the analysis on natural phenotypes are performed on a rather small set of genes, namely 10 and 65 genes respectively (while the total number of genes for *H. sapiens* surpasses 20,000). In the analysis of the natural phenotype, the 65 genes are chosen based on knowledge available in the literature. Therefore, the DL framework presented in this paper seems to have been designed for investigating phenotypes that have already been studied by means of methods capable of detecting linear associations (which is also supported by the definition of your null hypothesis for statistical evaluation of your findings, e.g. "the null hypothesis here is that genes only have main effects on the phenotype and no interactions"). Therefore, this should be made clear in the Abstract, in the Introduction, and in the Discussion. For example, when referring to "all SNPs" in the abstract, one could think of all the SNPs across the genome for a certain organism, while in the manuscript "all SNPs" refers to the SNPs that can be mapped onto the subset composed of the 10 or 65 selected genes. This should be clarified/detailed. Similar argument applies to "all gene-gene interactions". Furthermore, I would add a discussion about the possible advantages as well as drawbacks of this characteristic of your method.

2) I found the three different simulation settings appropriate for the assessment of different performance's aspects of your method. Also considering different signal-to-noise-ratios is a good experiment. Here some comments and suggestions for further improving the simulations:

2.1) Simulating that all the SNPs in the gene are causal for the simulated phenotype is a bit unrealistic. I would add at least 2 different settings where you have only a subset of the SNPs contributing to the phenotype, e.g. 1% and 10%, or 10% and 50%.

2.2) Another parameter you could vary in the simulations is the percentage of signal coming from the "main effect" and the "interactions effect". Now it is set at 80:20, but you could explore other settings, just to assess whether your method is stable under different conditions.

2.3) Have you considered including genes that, despite having a “main effect”, do not present interactions? Suppose you have a pool of 200 genes showing main effects with a phenotype of interest. It’s realistic to think that maybe only a subset of those presents interactions. What I would suggest is to add such an experiment, namely you could still consider 10 genes having main effects, but to simulate interactions only between (1) 5 of them and (2) 2 of them, for example.

2.4) Have you tried to increase the order of magnitude of the number of genes considered in your simulations? I think for some natural phenotypes it would be realistic to have more than 100 genes presenting main effects. Since the number of possible interactions in such a case increases compared to 10, it would be interesting to explore the robustness of your method in such conditions.

Note that 2.1, 2.2, 2.3, 2.4 hold for the 2/3 simulation settings, e.g. complex interactions and simple interactions.

2.5) Why do you include 2 out of 6 baselines in Fig 2.a? I would include the performance of all of them, at least in the Supplementary material.

3) In the following, some advice for improvement concerning the p-value calculation and the evaluation method:

3.1) Since the number of simulated causal interactions (6) is much lower compared to the total number of interactions, i.e. there is unbalance between positive interactions (e.g. interactions that are simulated as causal for the phenotype) and negative interactions (e.g. interactions that are not simulated as causal for the phenotype), I would recommend to use precision-recall curve instead of ROC curve, as precision-recall curve is more adapt to evaluate performance in such a scenario (different number of interactions simulated compared to the total number of possible interactions).

3.2) Figure 1.a, are you considering the maxT or the FDR way of constructing the null distribution (and therefore to obtain the p-values)? It should be stated in the caption. Moreover, if it’s the FDR way of constructing the null distribution, you should apply Benjamini-Hochberg to detect interactions, with a defined FDR threshold (usually 0.1 is used); if it’s maxT, then thresholding is statistically correct, but I would still precise which null distribution you use in the caption. As a minor comment, why are you representing only these 3 true and 3 false interactions? Are these interactions different from the others somehow? Why did you choose them?

3.3) Figure 5, bottom: the p-values here seem to be deflated, e.g. the maxT way of obtaining p-values is leading to p-values that are way too conservative, e.g., this approach is not leading to calibrated p-values. I wouldn’t consider this as a possible way of constructing the null distribution as the p-values appear not to be calibrated. I would recommend to show these results in the appendix and, then, use the other method to derive the null distribution and then the p-values (and consider it as the default one).

4) Comments on the experiments on natural phenotype:

4.1) The analysis is done on one phenotype only, e.g. phenotype HDL. What would be interesting and, I believe, necessary is to use the proposed method on other phenotypes as well. At least other 2/3 phenotypes/case study (more if it is possible).

4.2) It’s really positive that you have another dataset to verify your findings. However, I would not include results presenting FDR greater than 0.1, which is the FDR threshold that is often used in practice (which is already quite high as it allows the 10% of the findings to be FPs).

Minor comments:

- i) Could you detail more what you mean with "preferences to certain kinds of functions" in line 69 on page 3, and "The architecture reflects the domain knowledge" line 73 in the same page?
- ii) I found it a bit misleading referring to the effect/contribution of a gene on/to the phenotype as "gene expression" or saying "gene expresses"; this, in fact, has a defined meaning in bioinformatics. I would suggest either to change these expressions or to clearly state what you mean by "gene expression"/"gene expresses" in your manuscript. I would lean towards the first option.
- iii) Figure 1 in the Supplementary material: I would make the caption self-contained, e.g. I would detail that for the UK Biobank you used 65 genes and the relative SNPs. Moreover, when you say "we did not observe much high correlation" I would give an order of magnitude.
- iv) d_i in equation (1) is not defined.
- v) It might be a bit misleading to refer to the ground truth on the simulated interactions as "true" and "false", e.g. in line 129 of page 5. I would suggest a table (maybe in the Supplementary) where you collect these definitions.
- vi) Supplementary material, figure 4: I would label that the different figures are from different sizes of the data directly in the figure.
- vii) In some part of the paper, inline enumerators are in 1) format, while in others are 1. I would choose one way of formatting and be consistent.
- viii) page 7, line 164–165: "Note that selecting genes based on their main effects does not bias the assessment of the significance of interactions using permutation." -> I would add a citation that supports this statement.
- ix) it's great that the code is available. Make sure you declare the software license.
- x) Figure 8 is blurred.

Reviewer #2 (Remarks to the Author):

Authors in this paper have proposed a method for prediction of interactions among SNPs. They have proposed to use neural networks to capture the relations among SNPs and shapely score to quantify them. The method is tested on simulated datasets and independent sets as well.

Major Concerns:

- 1) The nn architecture is not clearly specified in the paper. It is not clear how the SNP's and gene interactions are encoded. In the main effect network, there are no hidden layers after the gene layer. What is the possible reason or intuition of such design consideration?
- 2) The nature of the hidden layers (before and after gene layer, number of layers, neurons, connections) should be elaborated along with gene layer specifications? How is it ensured that the "gene layer" is restricted to act like one?
- 3) How is that layer working to reflect the gene interactions? Is that dependent on the number of hidden layers? What are the inter-layer connections in the gene layer?

4) To create the training dataset, was any redundancy test performed? please specify the redundancy parameters and the details of the method.

5) Interactions between genes are already addressed in the literature [1,2] to predict phenotypes, i.e. the target variable. Thus the baseline used in this paper for comparison without considering any interactions i.e. the null model is a weak one and make the claim of the contribution a bit weaker to this reviewer. Please justify the claim with more ablation experiments.

[1] Costanzo, Michael; Kuzmin, Elena; van Leeuwen, Jolanda; Mair, Barbara; Moffat, Jason; Boone, Charles; Andrews, Brenda (2019). Global Genetic Networks and the Genotype-to-Phenotype Relationship. *Cell*, 177(1), 85–100. doi:10.1016/j.cell.2019.01.033

[2] Kuzmin, Elena, Benjamin VanderSluis, Wen Wang, Guihong Tan, Raamesh Deshpande, Yiqun Chen, Matej Usaj et al. "Systematic analysis of complex genetic interactions." *Science* 360, no. 6386 (2018): eaao1729.

Response to Reviewers for article “Gene-Gene Interaction Detection with Deep Learning”

We thank the reviewers and editors for their insightful comments on this manuscript. We have taken into account all the suggestions, as detailed below, and have highlighted the corresponding changes in the manuscript.

Reviewer #1:

The paper entitled “Gene-Gene Interaction Detection with Deep Learning” presents a Deep Learning (DL) approach to identify interactions between pairs of genes given a phenotype and a set of genes related to the phenotype under study. In particular, they design a Neural Network architecture presenting structured sparsity that allows to learn gene representations from the SNPs and, in a deeper layer, learns complex interactions, quantified by calculating the Shapley interaction score, between the genes and the phenotype. Moreover, the authors propose a permutation technique to assess the statistical significance of the detected interactions.

Overall impression of the work

The paper is overall very well written and well structured. The method is interesting and timely as there is growing interest in expanding GWAS tools to be more comprehensive and capable of detecting interactions at SNP level and/or gene level. Moreover, learning gene representations from the SNPs is a very interesting and valuable aspect of the work, since the gene representations that are available/used nowadays in the literature often seem to be overly simplistic (e.g., top associated SNP to represent a gene). However, there are few aspects of the method and its objectives that should be clarified; more details in the subsequent lines. The simulations and analysis on natural phenotype are quite comprehensive, some suggestions for improvement in the following. Furthermore, some advice for improvement concerning the evaluation framework are reported as well.

Specific comments:

1. Both the simulations and the analysis on natural phenotypes are performed on a rather small set of genes, namely 10 and 65 genes respectively (while the total number of genes for *H. sapiens* surpasses 20,000). In the analysis of the natural phenotype, the 65 genes are chosen based on knowledge available in the literature. Therefore, the DL framework presented in this paper seems to have been designed for investigating phenotypes that have already been studied by means of methods capable of detecting linear associations (which is also supported by the definition of your null hypothesis for statistical evaluation of your findings, e.g. “the null hypothesis here is that genes only have main effects on the phenotype and no interactions”). Therefore, this should be made clear in the Abstract, in the Introduction, and in the Discussion. For example, when referring to “all SNPs” in the abstract, one could think of all the SNPs across the genome for a certain organism, while in the manuscript “all SNPs” refers to the SNPs that can be mapped onto the subset composed of the 10 or 65 selected genes. This should be clarified/detailed. Similar argument applies to “all gene-gene interactions”. Furthermore, I would add a discussion about the possible advantages as well as drawbacks of this characteristic of your method.

Response: *We agree that the proposed method is designed to detect gene-gene interactions between a small candidate set of genes that have already been associated with the phenotype, instead of the set of all human genes. This is because neural networks (NNs) with common architectures are computationally infeasible to use with genome-wide SNP data due to the extremely high dimension. Possible solutions including devising tailored NNs architectures for the whole genome or applying distributed NNs. We clarify the suitable use cases of the method in the manuscript (see abstract, line 19 and line 234 in the main text) and discuss its advantages and limitations (line 250-254 in the discussion).*

2. I found the three different simulation settings appropriate for the assessment of different performance’s aspects of your method. Also considering different signal-to-noise-ratios is a good experiment. Here some comments and suggestions for further improving the simulations:

2.1. Simulating that all the SNPs in the gene are causal for the simulated phenotype is a bit unrealistic. I would add at least 2 different settings where you have only a subset of the SNPs contributing to the phenotype, e.g. 1% and 10%, or 10% and 50%.

Response: *We extend our experiments by simulating phenotypes with 10%, 50%, and 100% of total SNPs being causal (see line 119 in Section 2.2.1), and we include the results in Figure 3. We observe that NNs are better than existing approaches in all three settings, but the improvement of NNs over top-SNP methods becomes smaller when fewer SNPs are causal and the top-SNP assumption is more accurate (see line 151-154).*

2.2. Another parameter you could vary in the simulations is the percentage of signal coming from the “main effect” and the “interactions effect”. Now it is set at 80:20, but you could explore other settings, just to assess whether your method is stable under different conditions.

Response: *We simulate phenotypes with different main-interaction-ratios (M/I s, defined by the ratio between the variance of the main effect and the interaction effect): 20:80, 50:50, and 80:20, in Section 2.2.1 (line 119, and results are shown in Figure 3). We observe that our conclusions are consistent under different conditions, and using NNs are especially beneficial when the interaction effects dominate.*

2.3. Have you considered including genes that, despite having a “main effect”, do not present interactions? Suppose you have a pool of 200 genes showing main effects with a phenotype of interest. It’s realistic to think that maybe only a subset of those presents interactions. What I would suggest is to add such an experiment, namely you could still consider 10 genes having main effects, but to simulate interactions only between (1) 5 of them and (2) 2 of them, for example.

Response: *We add simulation experiments with only 2 and 5 interacting genes out of 10 genes in the simulator. We include the detailed experimental setup and results in the Supplementary Section 2.2 and refer to them in line 155 of the main text. We observe that NNs can detect interactions even more accurate than existing methods in these cases.*

2.4. Have you tried to increase the order of magnitude of the number of genes considered in your simulations? I think for some natural phenotypes it would be realistic to have more than 100 genes presenting main effects. Since the number of possible interactions in such a case increases compared to 10, it would be interesting to explore the robustness of your method in such conditions. Note that 2.1, 2.2, 2.3, 2.4 hold for the 2/3 simulation settings, e.g. complex interactions and simple interactions.

Response: *We add simulation experiments with 50 and 100 genes in the simulator. We include the detailed experimental setup and results in the Supplementary Section 2.3, referred to in line 156 of the main text. We observe that as the number of genes of the simulator increases, the performance of all methods drops (especially when the number of genes becomes 100), but NNs can still detect interactions better.*

As a summary, results of 2.1 and 2.2 are shown in Figure 3 (complex interactions) and Figure 4 (simple interactions) with description in line 113-120, line 149-157, and line 162-171 of the main text, and experiments of 2.3 and 2.4 are shown in Supplementary Section 2.2 and Supplementary Section 2.3 (complex interactions) and Supplementary Section 3.1 and Supplementary Section 3.2 (simple interactions).

2.5. Why do you include 2 out of 6 baselines in Fig 2.a? I would include the performance of all of them, at least in the Supplementary material.

Response: *We include all 6 baselines as well as the NN approach in the updated Figure 2a, as described in line 123-129. Moreover, we decided to remove the null distributions of interactions of Lasso and XGB from Figure 2b for clarity, because we focus on the null distribution of interactions of NNs in this work.*

3. In the following, some advice for improvement concerning the p-value calculation and the evaluation method:

3.1. Since the number of simulated causal interactions (6) is much lower compared to the total number of interactions, i.e. there is unbalance between positive interactions (e.g. interactions that are simulated as causal for the phenotype) and negative interactions (e.g. interactions that are not simulated as causal for the phenotype), I would recommend to use precision-recall curve instead of ROC curve, as precision-recall curve is more adapt to evaluate performance in such a scenario (different number of interactions simulated compared to the total number of possible interactions).

Response: We now use both precision-recall curve (PRC) and ROC curve to evaluate methods (see line 127-129). Specifically, we report only summary statistics of PRC and ROC, e.g., averaged precision (AP) and AUROC, in the main text (Figure 2 and Figure 3), and show the curves in Supplementary Section 2.4 and 2.5.

3.2. Figure 1.a, are you considering the maxT or the FDR way of constructing the null distribution (and therefore to obtain the p-values)? It should be stated in the caption. Moreover, if it's the FDR way of constructing the null distribution, you should apply Benjamini-Hochberg to detect interactions, with a defined FDR threshold (usually 0.1 is used); if it's maxT, then thresholding is statistically correct, but I would still precise which null distribution you use in the caption. As a minor comment, why are you representing only these 3 true and 3 false interactions? Are these interactions different from the others somehow? Why did you choose them?

Response: We use maxT to construct the null distribution in Figure 2a, and we specify the construction method in the updated caption. We randomly chose 3 true and 3 false interactions previously, but now we show all interactions in the updated Figure 2b.

3.3. Figure 5, bottom: the p-values here seem to be deflated, e.g. the maxT way of obtaining p-values is leading to p-values that are way too conservative, e.g., this approach is not leading to calibrated p-values. I wouldn't consider this as a possible way of constructing the null distribution as the p-values appear not to be calibrated. I would recommend to show these results in the appendix and, then, use the other method to derive the null distribution and then the p-values (and consider it as the default one).

Response: We agree and we move the maxT way to the Supplementary Section 4.2.

4. Comments on the experiments on natural phenotype:

4.1. The analysis is done on one phenotype only, e.g. phenotype HDL. What would be interesting and, I believe, necessary is to use the proposed method on other phenotypes as well. At least other 2/3 phenotypes/case study (more if it is possible).

Response: We now consider another two cholesterol phenotypes: low-density lipoprotein (LDL) and total cholesterol (TC) in real-world datasets, because they are measured on both UK Biobank and FINRISK datasets. We update the experimental setup in line 187-188, data preprocessing in line 273, 275, and 282, and show results on 3 phenotypes on the updated Figure 6, Figure 7, and Table 1. We update the result analyses in line 207-209, line 214-219, and line 222-229.

4.2. It's really positive that you have another dataset to verify your findings. However, I would not include results presenting FDR greater than 0.1, which is the FDR threshold that is often used in practice (which is already quite high as it allows the 10% of the findings to be FPs).

Response: Now we set the FDR threshold to 0.1, please see line 193 and updated Table 1 for more detail. While redoing the analysis with the updated threshold, we also change the NN regularization from the variational inference to the L_1 regularization with SGD training (see line 307-311 in Methods), because recent work shows that NNs with variational inference may underfit the data seriously [Tomczak et al., 2021]. We update both simulation and real-world experimental results accordingly, and we notice that with $FDR < 0.1$, we can detect 6 interactions for the HDL phenotype (see updated Figure 6 and Table 1), while the previous version can only detect 2 interactions.

Minor comments:

i) Could you detail more what you mean with "preferences to certain kinds of functions" in line 69 on page 3, and "The architecture reflects the domain knowledge" line 73 in the same page?

Response: This means that if we have domain knowledge about what types of functions that NNs should learn, we could design the NN architecture accordingly. For example, convolutional and pooling layers are used heavily in computer vision, because images are translation invariant in image classification tasks. See line 68 and 77.

ii) I found it a bit misleading referring to the effect/contribution of a gene on/to the phenotype as "gene expression" or saying "gene expresses"; this, in fact, has a defined meaning in bioinformatics. I would suggest either to change these expressions or to clearly state what you mean by "gene expression"/"gene expresses" in your manuscript. I would lean towards the first option.

Response: We change "how the gene expresses" to "how the gene behaves" for clarification. See line 79, 293.

iii) Figure 1 in the Supplementary material: I would make the caption self-contained, e.g. I would detail that for the UK Biobank you used 65 genes and the relative SNPs. Moreover, when you say “we did not observe much high correlation” I would give an order of magnitude.

Response: *We update the caption accordingly.*

iv) d_i in equation (1) is not defined.

Response: *g_i in equation (1) was actually defined before the equation.*

v) It might be a bit misleading to refer to the ground truth on the simulated interactions as “true” and “false”, e.g. in line 129 of page 5. I would suggest a table (maybe in the Supplementary) where you collect these definitions.

Response: *We define “true” and “false” interactions in line 112.*

vi) Supplementary material, figure 4: I would label that the different figures are from different sizes of the data directly in the figure.

Response: *We label the different configurations of different figures in the figure. Please see Supplementary Section 2.4 and 2.5.*

vii) In some part of the paper, inline enumerators are in 1) format, while in others are 1. I would choose one way of formatting and be consistent.

Response: *We change 1) format to 1. format. See line 41 and line 234.*

viii) page 7, line 164–165: “Note that selecting genes based on their main effects does not bias the assessment of the significance of interactions using permutation.” - I would add a citation that supports this statement.

Response: *This is one observation from the permutation study in Section 2.2.3, where the permutation distribution of interactions is well-calibrated with the existence of main effects. See line 185.*

ix) it’s great that the code is available. Make sure you declare the software license.

Response: *We declare the license in line 374.*

x) Figure 8 is blurred.

Response: *We change Figure 8 to pdf format.*

Reviewer #2:

Authors in this paper have proposed a method for prediction of interactions among SNPs. They have proposed to use neural networks to capture the relations among SNPs and shapely score to quantify them. The method is tested on simulated datasets and independent sets as well.

Major concerns:

1. The nn architecture is not clearly specified in the paper. It is not clear how the SNP’s and gene interactions are encoded. In the main effect network, there are no hidden layers after the gene layer. What is the possible reason or intuition of such design consideration?

Response: *The NN architecture is shown in updated Figure 1 where the hidden layers are fully connected feed-forward networks (Multilayer Perceptron, MLP). SNPs are represented by the centered number of minor alleles, and genes are represented by the values of “gene nodes”. The interactions between SNPs and genes are implicitly encoded into the MLPs of the gene interaction NN **after the network is trained on the data**, because an MLP can approximate any function (including interactions) according to the universal approximation theorem. And we use gene-gene interaction score (introduced in Section 2.1) to reveal these interactions from the trained gene interaction NN. We don’t use architectures that explicitly specify the interaction between genes as this could limit the interactions that the model can detect. See line 70-76 for detailed explanation.*

The main effect neural network (NN) is used to learn the main effects of genes separately from the interaction effects. This allows creating permuted datasets where the main effect is intact but the interaction effects are removed by permutation. This is because the null hypothesis of interaction detection is that the interaction effect does not exist but the main effect may exist, and we showed that the new permutation method with the main effect NN is well-calibrated in Section 2.2.3 where existing permutation methods are not. The permutation test is used to construct null distributions for interaction scores in order to control statistical errors, e.g. by using the false discovery rate (FDR) in Section 2.3.

2. The nature of the hidden layers (before and after gene layer, number of layers, neurons, connections) should be elaborated along with gene layer specifications? How is it ensured that the “gene layer” is restricted to act like one?

Response: We provide the detailed configuration of neural network architectures in line 304-307 in the Method section. The “gene-layer” is restricted by the sparse architecture such that each node in the gene-layer can only receive information from the SNPs within the corresponding gene.

3. How is that layer working to reflect the gene interactions? Is that dependent on the number of hidden layers? What are the inter-layer connections in the gene layer?

Response: Feed-forward neural networks with at least one non-linear hidden layer can approximate any interactions in the data according to the universal approximation theorem (please check the response to 1). Therefore, the gene interaction NN does not use any inter-layer connections (e.g., between gene nodes) to learn interactions, but rather the interactions are learned by the subsequent fully connected layers. We remove the blue arrows between nodes in the “gene layers” for clarification (see updated Figure 1). Graph neural networks could have inter-layer connections to encode known interactions between genes, but such specific architecture prevents the user discovering new interactions from the data (see line 256-258 of Discussion).

4. To create the training dataset, was any redundancy test performed? please specify the redundancy parameters and the details of the method.

Response: To create the training datasets, we have additional quality control and data preprocessing steps. We use the same methods and hyper-parameters for these two steps as previous works, described in the Data preprocessing part of the Methods section. Moreover, the 65 candidate genes that we used in the paper are selected from previous GWAS results on cholesterol phenotypes. See the Data preprocessing section.

5. Interactions between genes are already addressed in the literature [1,2] to predict phenotypes, i.e. the target variable. Thus the baseline used in this paper for comparison without considering any interactions i.e. the null model is a weak one and make the claim of the contribution a bit weaker to this reviewer. Please justify the claim with more ablation experiments.

1. Costanzo, Michael; Kuzmin, Elena; van Leeuwen, Jolanda; Mair, Barbara; Moffat, Jason; Boone, Charles; Andrews, Brenda (2019). Global Genetic Networks and the Genotype-to-Phenotype Relationship. *Cell*, 177(1), 85–100. doi:10.1016/j.cell.2019.01.033

2. Kuzmin, Elena, Benjamin VanderSluis, Wen Wang, Guihong Tan, Raamesh Deshpande, Yiqun Chen, Matej Usaj et al. ”Systematic analysis of complex genetic interactions.” *Science* 360, no. 6386 (2018): eaao1729.

Response: The purpose of this paper is to provide a data-driven/statistical approach to detect complex pairwise interactions between genes from GWAS datasets with deep learning, instead of using experimental approaches to construct genetic networks (including higher-order interactions) discussed in [1,2], but how to use existing genetic networks to increase statistical power is an interesting direction and we add a discussion in line 256-258. The null model, i.e., main effect NN, serves to construct calibrated null distributions of interactions to assess their statistical significance, instead of a baseline. Explicitly including existing interactions in the null model, e.g. by concatenating genes that are known to interact, would require extending the model which, while an interesting topic, falls outside the scope of the current work, and we note that the permutation procedure with current null model already yields good results (see Section 2.2.3). However, we expect that incorporating information about known interactions into the null model could improve the accuracy of detecting novel interactions further (or at least it should not decrease the accuracy); hence our results can be seen as providing a lower bound for performance of the more elaborate case when existing interactions are explicitly accounted for.

To summarize, we provide 6 baselines for interaction detection in the simulation (see line 121-123), and NNs outperform all of them when interactions are complex (see Figure 3). Moreover, in real-world analysis, NNs can detect replicable interactions that LR with multiplicative interaction (considered in the quantitative analysis of [2]) between top SNPs can ignore (see Table 1). In addition, we add new ablation studies to test how NNs

improve phenotype prediction with candidate interacting genes in line 229-232 and Supplementary Section 6.1. NNs improve LR with multiplicative interaction model in terms of test R^2 by 36.7% if the top SNPs are used in LR to represent genes (i.e., the existing approach) and by 1.3% if gene representations learned from NNs are used. This indicates that it's beneficial to consider all possible SNPs from corresponding genes, as well as their complex interactions.

References

M. Tomczak, S. Swaroop, A. Foong, and R. Turner. Collapsed variational bounds for bayesian neural networks. *Advances in Neural Information Processing Systems*, 34:25412–25426, 2021.

Reviewers' comments:

Reviewer #1 (Remarks to the Author):

All my previous comments have been well addressed by the authors, thank you.

I have one last minor question about the learning framework. Is the validation scheme the same for the simulations and the real world data application? I.e., the details in lines 190–191, "We split the preprocessed data into 70% for training and 30% for testing." belongs to the Section on the application on the real data, while for the simulations this aspect is unclear. Furthermore, the number of folds of the cross-validation (lines 308-310) should be specified.

Reviewer #2 (Remarks to the Author):

The authors have revised the manuscript by addressing several points. This reviewer has one concern left and that is the comparison of NN with traditional models like LR and XGB. That ablation part is still weak and fails to provide any explanatory information or knowledge about the success of the proposed model.

Response to Reviewers for article “Gene-Gene Interaction Detection with Deep Learning”

We thank the reviewers for their insightful comments on this manuscript. We have taken into account all the suggestions, as detailed below, and have highlighted the corresponding changes in the manuscript.

Reviewer #1:

All my previous comments have been well addressed by the authors, thank you.

I have one last minor question about the learning framework. Is the validation scheme the same for the simulations and the real world data application? I.e., the details in lines 190–191, “We split the preprocessed data into 70% for training and 30% for testing.” belongs to the Section on the application on the real data, while for the simulations this aspect is unclear. Furthermore, the number of folds of the cross-validation (lines 308-310) should be specified.

Response: *We use the same validation scheme for both simulation and real world data, and we now mention this in the data preprocessing part of the Method section (line 277). We use 5-fold cross-validation for hyperparameter selection in all settings (see line 311 in the Method section).*

Reviewer #2:

The authors have revised the manuscript by addressing several points. This reviewer has one concern left and that is the comparison of NN with traditional models like LR and XGB. That ablation part is still weak and fails to provide any explanatory information or knowledge about the success of the proposed model.

Response: *We have added ablation studies on two simulated datasets with complex interactions to separate the benefit of learning gene representations and modeling complex genetic interactions in the proposed method (Supplementary Section 2.6). We mention now the ablation studies in the main text accordingly (line 157 in Section 2.2.1). We observe that learning gene representations from the corresponding SNPs always improves interaction detection, especially when the number of causal SNPs is large. Moreover, modeling complex interactions between gene representations with NNs can further improve their detection. We also observe that simply using a NN on the top-SNP representations does not work properly, and has a similar performance to the XGBoost model applied on the top SNPs. This highlights the importance of using more continuous gene representations instead of the Binomially distributed top-SNP representations when complex models are used to learn genetic interactions.*